# WHICH PRE-TRAINED MODEL IS EFFECTIVE FOR SPEECH SEPARATION ?

## ABSTRACT

The effectiveness of the use of general audio pre-trained models to generate representations suitable for speech separation has been explored in a previous study (Huang et al., 2022) with the main finding being that they provide minimal benefit compared to features extracted without the models. The study hypothesised that since the general audio pre-trained models were trained with clean audio dataset, they are unable to generalize to noisy and mixed speeches hence not effective in speech separation. This paper investigates this hypothesis by comparing the performance of pre-trained model trained on contaminated speeches and that trained on clean ones. We are interested in evaluating whether contamination leads to better downstream performance. We also investigate if the type of input used to train the pre-trained model impacts the quality of embeddings it generates. To separate the sources, we propose a fully unsupervised technique of speech separation based on deep modularization. Our findings establish that by injecting noise and reverberation in the training dataset, the pre-trained model generate significantly better embeddings than when clean dataset is used. Further, based on the model presented here, working in short-time Fourier transform (STFT) results in better features than using time-domain features. The proposed deep modularization speech separation technique can improve SI-SNRi and SDRi by 1.3 and 2.7, respectively, when mixtures contain less than four sources and improves the results significantly for many source mixtures.

## 1 INTRODUCTION

Pre-trained models have become popular especially in natural language processing (NLP) and Computer vision. In NLP, for example, a large corpus of text can be used to learn universal language representations which are beneficial for downstream NLP tasks. Due to their success in these domains, unsupervised learning-based pretrained models have been introduced for audio data (Chung et al., 2019) (Chung et al., 2020) (Liu et al., 2020) (Liu et al., 2021)(Baevski et al., 2020)(Hsu et al., 2021a). Such pretrained models can be beneficial to speech separation in several ways: First, the models are trained using large speech dataset hence can learn universal speech representations which can boost the quality of speech generated by speech separation models. Secondly, they provide models with better initialization which can result in better generalization and speed up convergence during training of speech separation models. Finally, pretrained speech models can act as regularizers to help speech separation models to avoid overfitting (Erhan et al., 2010). The importance of pre-trained models in speech separation is the subject of investigation in (Huang et al., 2022). They use 13 speech pre-trained models to generate features of a mixed speech which are then passed through a three-layer BLSTM network to generate speech separation mask. They compare the performance of these features with those of baseline STFT and Mel filterbank (FBANK) features. Their experiments establish that the 13 pre-trained models used do not significantly improve feature representations as compared to those of baselines. Hence the quality of separated speech generated based on features of pre-trained models are only slightly better or worse in some cases as compared to those generated based on the baseline features. They attribute this to domain mismatch and information loss. Since most of the pre-trained models were trained with clean speech, they do not generalize well to noisy speech domain. Pre-trained models are usually trained to model global features and long-term dependencies hence some local features of the noisy or mixed speech signal may be lost due to this during feature extraction. Using HuBERT large model (Hsu et al., 2021b),

they demonstrate that the last layer of the model does not produce the best feature representation for speech enhancement and separation. In fact, for speech separation, the higher layers features are of low quality as compared to lower layers. They show that the weighted-sum representations of the representations from the different layers of pre-trained models where lower layers are given more weight generate better speech the enhancement and separation results as compared to isolated layers representations. They hypothesise that this could be due to the loss of some local signal information necessary for speech reconstruction tasks in deeper layers. In this research, we re-look into the use of pre-trained models to boost speech separation. We are particularly investigating whether training a pre-trained model on contaminated speeches will result in it generating quality features for speech separation as compared to that trained with clean speech. We also investigate whether input features (Fourier vs time domain features) have a significant impact on the quality of features generated by the pre-trained model. Another major challenge faced by DNN models performing speech separation is the permutation ambiguity. Most speech separation tools such as (Zeghidour & Grangier, 2021a) (Huang et al., 2011) (Weng et al., 2015) (Isik et al., 2016) (Hershey et al., 2016a) and (Luo & Mesgarani, 2019a) cast the problem of speech separation as a multi-class regression. In that case, training a DNN model involves comparing its output to a source speaker. The models always output a dimension for each target class and when multiple sources of the same type exist, the system needs to select arbitrarily which output dimension to map to each output and this raises a permutation problem (permutation ambiguity) Hershey et al. (2016a). Systems that perform speaker separation have an extra burden of designing mechanisms that are geared towards handling the permutation problem. One key technique of avoiding permutation ambiguity is to perform speech separation through clustering technique (Hershey et al., 2016b) (Byun & Shin, 2021) (Isik et al., 2016) (Qin et al., 2020) (Lee et al., 2022). Despite their success, the existing clustering technique employ supervised training which require a costly process of data labelling. Furthermore, these methods require that the number of speakers need to be known before execution which may not be practical in some cases. We seek to avoid permutation ambiguity problem by implementing a fully unsupervised speech separation technique in the downstream using deep modularization network where the number of speakers need not to be known priori. In summary we make the following key contributions:

1. We show that training a pre-trained model with contaminated audio generates better features talored for speech separation as compared to that trained on clean audio.

2. We show that the type of input (DFT transformed input vs raw waveform) used to train a model has significant effect on the quality of features the model generates.

3. We propose a new fully unsupervised technique for speech separation to avoid permutation problem. The proposed technique is able to scale to mixtures with many speakers with only small drop in performance.

## 2 RELATED WORK

Speech enhancement and separation tools can be categorised into two broad categories based on the type of input features, i.e., those using Fourier spectrum features as input and those using time domain features. Fourier spectrum-based features tools do not work directly on the raw signal (i.e., signal in the time domain) rather they incorporate the discrete Fourier transform (DFT) in their signal processing pipeline mostly as the first step to transform a time domain signal into frequency domain. These models recognise that speech signals are highly non-stationary, and their features vary in both time and frequency. These features include Log-power spectrum features (Fu et al., 2017) (Du & Huo, 2008) (Xu et al., 2015) (Du et al., 2014), Mel-frequency spectrum features (Liu et al., 2022) (Ueda et al., 2016) (Du et al., 2020) (Fu et al., 2018) (Weninger et al., 2014) (Donahue et al., 2018), DFT magnitude features (Nossier et al., 2020) (Fu et al., 2018) Grais & Plumbley (2018) Fu et al. (2019) Jansson et al. (2017) Kim & Smaragdis (2015) and Complex DFT features (Fu et al., 2017) (Williamson & Wang, 2017) (Kothapally & Hansen, 2022a) (Kothapally & Hansen, 2022b). The assumption made by most DNN models that use Fourier spectrum features is that phase information is not crucial for human auditory. Therefore, they exploit only the magnitude or power of the input speech to train the DNN models to learn the magnitude spectrum of the clean signal and the factor in the phase during the reconstruction of the signal (Xu et al., 2014) (Kumar & Florencio, 2016) (Du & Huo, 2008) (Tu et al., 2014) (Li et al., 2017). The use of the phase from the noisy signal to

estimate the clean signal is based on works such as (Ephraim & Malah, 1984) that demonstrated that the optimal estimator of the clean signal is the phase of the noisy signal. Furthermore, most speech separation models work on frames that are of size between 20-40 ms and believe that the short-time phase contains low information (Lim & Oppenheim, 1979) (Oppenheim & Lim, 1981) (Vary & Eurasip, 1985) (Wang & Lim, 1982) and therefore are not crucial when estimating clean speech. However, recent research Paliwal et al. (2011) have demonstrated through experiments that further improvements in quality of estimated clean speech can be attained by processing both the short-time phase and magnitude spectra. Further, the factoring in of the noisy input phase during reconstruction has been noted to be a problem since the phase errors in the input interact with the amplitude of the estimated clean signal hence causing the amplitude of the estimated clean signal to differ with the amplitude of the actual clean signal being estimated (Erdogan et al., 2015), (Han et al., 2015). Due to phase challenge while working with Fourier spectrum features different tool such as (Luo & Mesgarani, 2018) (Luo et al., 2020) (Luo & Mesgarani, 2019a) (Venkataramani et al., 2018) (Zhang et al., 2020) (Subakan et al., 2021a) (Tzinis et al., 2020a) (Tzinis et al., 2020b)(Kong et al., 2022) (Su et al., 2020) (Lam et al., 2021b) (Lam et al., 2021a) explore the idea of designing a deep learning model for speech separation that accepts speech signal in the time-domain. The fundamental concept of these models is to replace the DFT-based input with a data-driven representation that is jointly learnt during model training. The models therefore accept as their input the mixed raw waveform sound and then generate either the estimated clean sources or masks that are applied on the noisy waveform to generate clean sources. By working on the raw waveform, these models address the key limitation of DFT-based models, since the models are designed to fully learn the magnitude and phase information of the input signal during training Luo et al. (2020). The DNN models for speech separation can also be categorised based on how they were trained, that is, spectral mapping techniques (Fu et al., 2018) (Grais & Plumbley, 2018) (Kim & Smaragdis, 2015) (Xu et al., 2015) (Lu et al., 2013) (Xu et al., 2014) (Fu et al., 2016) (Gao et al., 2016), spectral masking techniques Wang & Wang (2013) Isik et al. (2016) Weninger et al. (2014) Fu et al. (2016) Narayanan & Wang (2013) Chen et al. (2015) (Huang et al., 2015) (Hershey et al., 2016b) (Grais et al., 2014) (Zhang & Wang, 2016) (Narayanan & Wang, 2015) (Weninger et al., 2015) (Huang et al., 2011) Zhang & Wang (2016) Liu & Wang (2019a) and generative modelling(Donahue et al., 2018)(Fu et al., 2019). The use of pre-trained models in speech separation and enhancement has been explored in (Huang et al., 2022). Currently there is no pre-trained model trained specifically for speech separation, models exploit general audio pre-trained models in speech separation. However, their use has not resulted in significant performance boost. The work in (Huang et al., 2022) hypothesises that this could be because the pretrained models were not trained on noisy or contaminated speech; hence, they are unable to extrapolate to noisy speeches. This work investigates this hypothesis i.e., whether training a pre-trained model on contaminated speeches will result in it generating quality features for speech separation as compared to that trained with clean speech. We also investigate whether input features have an impact of the quality of features generated by the pre-trained model.

## 3    CONTRASTIVE DEEP MODULARIZATION MODEL (CONDEEPMOD)

### 3.1    FOURIER BASED FEATURE REPRESENTATION LEARNING WITHOUT AUGMENTATIONS

Here, we are interested in investigating whether features representations generated by speech frames where no explicit augmentations have been applied are ideal for speech separation. We implement the contrastive learning similar to the one proposed in Saeed et al. (2021) but at frame-level. The goal of contrastive self-supervised learning is to establish a representation function $f : x \mapsto R^d$ that maps augmentations to a $d-$dimensional vectors by ensuring that similar view of augmentations are closer to each other as compared to those of random ones. The practice is to pick augmentations $(x, x^+)$ that are obtained by passing a given input through two different augmentation functions. Ideal augmentations of inputs are those that retain features of the inputs that are crucial in the intended task (e.g., classification) but modify the features that are less important for that task. Here, we do not apply any explicit augmentation on the speech; we hypothesise that the different frames of a speech belonging to a given speaker qualify to be viewed as augmentations of a standard hidden frame of that speaker in the speech separation domain. For frames of a given speaker to qualify as augmented versions of each other, they must retain important features for speaker's speech identity and modify the less important ones. For speech separation, the key features are voice pitch, that is, auditory perception of the rate of vocal fold vibration (the fundamental frequency or F0) (Xie & Myers,

2015), vocal timbre, and speaking rate that reveal indexical characteristics can be used for talker identification. To investigate whether the different frames of a speaker are augmentations of each other with regard to speech separation, given a clean speech signal in the time domain $x \in R^T$, we transform the signal into a STFT representation $S \in R^{F \times T}$. From the resulting frequency-domain representations we extract frames of spectrogram (T-F bins).These frames of spectrograms serve as our datapoints. We then design a function $f : S \mapsto R^d$ that maps the frames to $d$-dimensional vectors by encouraging the representations of pairs of frames from a given speaker to be closer to each other than the representations of frames of another random speaker. Given $n$ speeches from $n$ speakers, we segment each of the $n$ STFT transformed speech signals into equally sized frames. Let $\bar{X}$ denote the set of all frames generated from the speeches $n$. Let the function $\mathcal{S}(., . \mid \bar{X})$ be viewed as an augmentation pair generator such that it picks two pairs of frames from $\bar{X}$ belonging to the same speaker, that is,

$$(x_i, x_i^+) \sim \mathcal{D}_{pos} \equiv (x_i, x_i^+) \sim \text{i.i.d } \mathcal{S}(., . \mid \bar{X}) \tag{1}$$

Here the pair $(x_i, x_i^+)$ is the positive pair with distribution $\mathcal{D}_{pos}$. Given a batch of size $b$, for a positive pair $(x_i, x_i^+)$, we consider all the other $b - 2$ to be the negative examples with a distribution of $\mathcal{D}_{neg}$. To train the model to fit the function $f$, we use adopt simCLR contrastive loss Chen et al. (2020).

$$\mathbb{E}_{x,x^+ \sim D_{pos}, x_{i:n-2}^- \sim D_{neg}} [-\log(\frac{e^{f(x)^T f(x^+)}}{e^{f(x)^T f(x^+)} + \sum_{i=1}^{n-2} e^{f(x)^T f(x_i^-)}})] \tag{2}$$

The loss function seeks to make the similarity $f(x)f(x^+)$ larger as compared to $f(x)f(x^-)$. Once the model is trained to establish frame level features, we exploit the trained model in the downstream task.

### 3.2 FOURIER BASED FEATURE REPRESENTATION LEARNING WITH AUGMENTATIONS

Here, given the set $\bar{X}$ defined in section 3.1, a randomly selected frame $x_i \in \bar{X}$ is augmented by applying two different augmentations functions $(a_1, a_2)$ to it to generate similar views i.e $a_1(x_i) \to \tilde{x}_i$ and $a_2(x_i) \to \tilde{x}_j$ (see figure 1). There are several potential enhancements that can be applied to speech,including pitch modification, additive noise, reverberation, band reject filtering, and time masking Kharitonov et al. (2021). In our case, we use additive noise $(a_1)$ and additive noise plus reverberation $(a_2)$. These two augmentations involve contaminating the selected frame by either adding noise or adding noise then convolving the noisy speech with impulse response. Noise is added to speech by adding selected nonstationary noises with a selected signal-noise ratio (SNR). With reverberation, impulse responses are used to simulate different acoustic conditions. Applying the augmentations on $b$ selected random frames from the set $\bar{X}$ results in 2b datapoints. For given positive pair $(\tilde{x}_i, \tilde{x}_j)$ within a batch, the other $2(b - 1)$ datapoints are treated as negative samples and loss function in equation 2 is used for training.

### 3.3 TIME DOMAIN-BASED FEATURE REPRESENTATION LEARNING WITHOUT AUGMENTATIONS

Given a time-domain speech signal $x \in R^T$, the signal is processed by an encoder similar to the one proposed in Subakan et al. (2021b) to generate representation $h \in R^{F \times T}$. This is then chunked into frames along the time axis to generate $L \in R^{F \times S \times N}$. Here $N$ represents the number of frames generated. Given $n$ speech signals, the resulting set $\bar{W}$ of $N \times n$ frames are then processed similar to $\bar{X}$ in the discussion in section 3.1.

### 3.4 TIME DOMAIN-BASED FEATURE REPRESENTATION LEARNING WITH AUGMENTATIONS

Given the set $\bar{W}$ established in section 3.3, a randomly selected frame $x_i \in \bar{S}$ is augmented by applying the two different augmentations functions $(a_1, a_2)$ defined in section 3.2 to it. This generates two similar views, i.e., $a_1(x_i) \to \tilde{x}_i$ and $a_1(x_i) \to \tilde{x}_j$. Once the augmentations have been generated, the processing proceeds similarly to the discussion in Section 3.2.

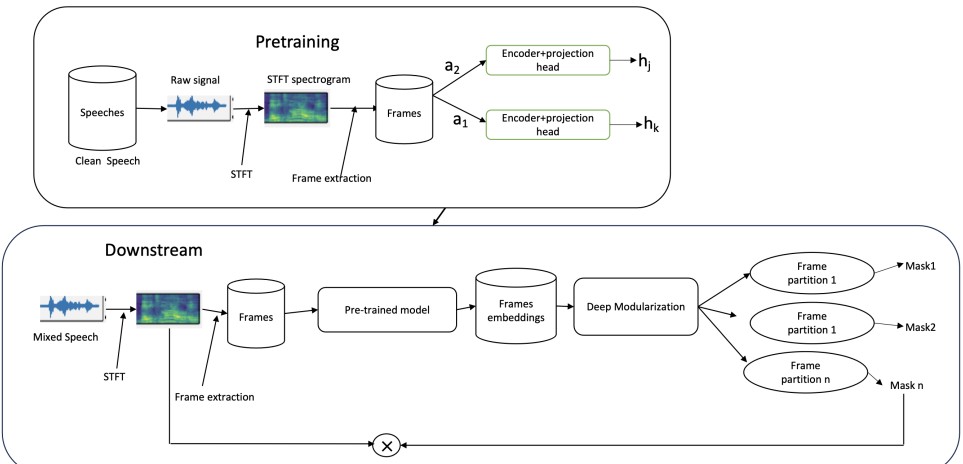

Figure 1: How the pre-trained model is trained when explicit augmentations are applied to STFT based frames. The lower part of the figure shows how the pretrained model is utilised for deep modularization. Speech reconstruction is not shown in the figure. $a_1$ and $a_2$ are the different augmentations

## 4 DOWNSTREAM TASK: FRAMES PARTITIONING

The goal is to exploit frame-level representation learned in section 3 to partition a set of frames such that frames dominated by a given speaker are grouped together. To proceed, we define a graph $G(V, E)$ where $V = (v_1, v_2, \cdots, v_n)$, $|V| = n$ is the set of all nodes (frames) and $E \subset V \times V$, $|E| = m$ is the set of all edges of the mixed speech signal. We denote the adjacency matrix of $G$ by $A$ where $A_{ij} = 1$ if $\{v_i, v_j\} \in E$ and 0 otherwise. The degree of $v_i$ is defined as $d_i = \sum_j^n A_{ij}$, we are interested in generating graph partition function $\mathcal{F} : V \to \{1, \cdots, k\}$ that splits the set of nodes $V$ into $k$ partition $v_i = \{v_j : \mathcal{F}(v_j) = i\}$ given the nodes attributes $\bar{F} \in R^{n \times d}$ generated by contrastive learning. In order to partition the vertices, we explore the statistical approach of vertices partitioning known as modularity (Q) (Newman, 2006). Modularity involves comparing the number of edges within partitions and some equivalent randomized partitions (null network) in which edges are placed without regard to relationships that exist in the network. Modularity is, therefore, defined as

$$Q = \text{Number of edges within partitions} - \text{expected number of such edges} \quad (3)$$

A high value of $Q$, indicates closer similarities among members belonging to a given partition. Therefore, the goal is to maximise $Q$. Modularity (Q) is derived in Newman (2006) as:

$$Q = \frac{1}{2m} \sum_{ij} (A_{ij} - P_{ij}) \delta(g_i, g_j) \quad (4)$$

where $\delta(g_i, g_j)$ is 1 if vertex $i$ and $j$ belong to the same partition and 0 otherwise. $P_{ij}$ is the expected number of edges between $i$ and $j$ while $A_{ij}$ is the actual number of edges between $i$ and $j$. If vertex $i$ and $j$ have degrees $d_i$ and $d_j$, respectively, then the expected degree of vertex $i$ can be defined as $\sum_j P_{ij} = d_i$. Based on this, vertex $i$ and $j$ are connected with probability $P_{ij} = \frac{d_i d_j}{2m}$ (see (Newman, 2006)). Hence equation 4 is modified to:

$$Q = \frac{1}{2m} \sum_{ij} (A_{ij} - \frac{d_i d_j}{2m}) \delta(g_i, g_j) \quad (5)$$

The problem of maximizing $Q$ is NP-Hard (Brandes et al., 2006), however, if we seek to generate $k$ non-overlapping partitions, a partition assignment matrix $S \in R^{n \times k}$ ($n$ represents number of vertices) is defined (Newman, 2006). Each column of $S$ indexes a partition, that is, $S = \{s_1 \mid s_2 \mid , \cdots, \mid s_k\}$. The columns are vectors of (0,1) elements such that $S_{ij} = 1$ if vertex $i$ belongs to partition $j$ and 0 otherwise. Based on this setup the columns of $S$ are mutually orthogonal since

each row of the matrix sums to 1. $S$ therefore satisfies the normalization $Tr(S^T S) = n$ where $Tr(.)$ is the matrix trace. Based on the definition of $S$, $\delta(g_i, g_j) = \sum_{k=1}^{k} S_{ik} S_{jk}$ and hence

$$Q = \frac{1}{2m} \sum_{ij=1}^{k} \sum_{n=1}^{k} (A_{ij} - P_{ij}) S_{ik} S_{jk} = \frac{1}{2m} Tr(S^T B S) \tag{6}$$

where $B$ is the modularity matrix such that $B_{ij} = A_{ij} - P_{ij}$. By relaxing $S \in R^{n \times k}$, the optimal $S$ that maximizes $Q$ is the top $k$ eigenvectors of matrix $B$. In our case, we seek to optimize $Q$( learn and optimize cluster assignment matrix $S$), by modularizing the frame features $\bar{F} \in R^{n \times d}$ learned via constrative learning. We seek to adapt the deep neural network graph partition technique proposed in Bianchi et al. (2020),Müller (2023) to partition our features. They partition nodes of a graph by the following formulation:

$$\bar{F} = \text{GNN}(\tilde{A}, X, \theta_{GNN}) \tag{7}$$

$$S = \text{softmax}(\bar{F}) \tag{8}$$

Where $\tilde{A} = D^{-\frac{1}{2}} A D^{-\frac{1}{2}}$, $X$ are the input features, $D$ is the diagonal matrix with the degrees $d_1, \cdots, d_n$ on the diagonal and $A$ is the adjacency matrix. In equation 7, node features $\bar{F}$ are learned via graph neural network (GNN) and the assignment matrix S is established via SoftMax activation function. In (Bianchi et al., 2020), the assignment matrix $S$ is established by multilayer perception (MLP) with SoftMax on the output layer. In our case, we formulate the problem as:

$$\bar{F} = \text{Con}(X, \theta_{con}) \tag{9}$$

$$S = \text{RNN}(\bar{F}, \theta_{rnn}) \tag{10}$$

Where the frame feature matrix $\bar{F}$ is established via contrastive learning (Con). The partition assignment of a frame is established using BLSTM similar to the one proposed in (Huang et al., 2022) with SoftMax on the output layer. This maps each frame feature $\bar{f}_i \in \bar{F}$ to the $i$ row of the cluster assignment matrix $S$. To optimise the assignment $S$, we use the loss function in Equation 12 (Müller, 2023). The loss is composed of a modularity (derived in Equation 6) term and a collapse regularizer. The collapse regularizer is crucial to avoid $S$ generating trivial partitions (Müller, 2023). Furthermore, it has been shown in Müller (2023) that the loss function in equation 12 maintains consistency of community detection as the number of nodes increases.

$$L(S) = -\frac{1}{2m} Tr(S^T B S) + \frac{\sqrt{k}}{n} \| \sum_i S_i^T \|_F - 1 \tag{11}$$

Here, $\|.\|_F$ is the Frobenius norm. The complexity of the modularity term $Tr(S^T B S)$ is $\mathcal{O}(n^2)$ per update of $L(S)$ which makes the training process computationally costly. Therefore, to efficiently update $L(S)$, Müller (2023) proposes to decompose $Tr(S^T B S)$ into sum of sparse matrix-matrix multiplication and rank one degree normalization $Tr(S^T A S - S d^T d S)$. This reduces the complexity to $(O)(d^2 n)$ for every update of the loss function.

$$L(S) = -\frac{1}{2m} Tr(S^T A S - S d^T d S) + \frac{\sqrt{k}}{n} \| \sum_i S_i^T \|_F - 1 \tag{12}$$

## 4.1 ADJACENCY MATRIX

To construct the adjacency matrix $A$, for each frame $i$ we compute its similarity with all other nodes using inner product i.e.

$$e_{ij} = \bar{f}_i^T \bar{f}_j \tag{13}$$

where $j = 1, 2, \cdots, n$ and $\bar{f}_i$ and $\bar{f}_j \in \bar{F}$. We then select a threshold $\theta$ such that if $e_{ij} < \theta$, we remove an edge between $i$ and $j$ then the adjacency matrix is defined as

$$A_{ij} = \begin{cases} 1, & \text{if there is an edge between } i \text{ and j} \\ 0, & \text{otherwise} \end{cases} \tag{14}$$

Optimum $\theta$ is established experimentally (explained in Appendix A3).

## 5 CLEAN SIGNAL ESTIMATION

From the established partitions $k$, we generate $k$ masks in the range $[0, 1]$, where 0 indicates that a given frame in the input mixed signal is missing in that cluster, while 1 signifies the presence of a given frame. The mask-based separation of sources is predicated on the assumption of sparsity and orthogonality of the sources in the mixed signal in the domain in which masks are computed. Based on this assumption, the dominant signal at a given range is taken to be only signal at that range. Therefore, the generation of clusters through modularization is used to estimate the dominant signals in a given range. Once the masks have been established, they are applied to the input mixed signal to generate $k$ estimated clean signals. For the input speech signal that has been transformed to STFT, the mask is applied to the input STFT spectrogram to obtain the estimated spectrograms of clean speech signals. The inverse STFT is then used to estimate a clean speech signal. In case of a time domain signal, the mask is applied to the STFT-like transformation generated by the encoder. The decoder (transposed encoder) is the used to generate estimated signal. For STFT phase reconstruction, we use the technique proposed in Wang et al. (2018) which jointly reconstructs the phase of all sources in each mixture by exploiting their estimated magnitudes and the noisy phase using the multiple input spectrogram inversion (MISI) algorithm (Gunawan & Sen, 2010).

### 5.1 MODEL(F)

The model (encoder) $f$ which is used to establish frame representations is made up of a stack of 6 identical layers. One such layer is shown in figure 2 (appendix A4). The layers are composed of layer normalisation, 1D convolution, and 1D maxpooling.

## 6 EXPERIMENTS

### 6.1 DATASET

To pre-train all the four model variants, we used the popular Wall Street Journal (WSJ0) corpus (Paul & Baker, 1992). The dataset was recorded using a close-talk microphone hence free from reverberation and noise. We used 30 hours of speeches from $si\_tr\_s$ to train the models. When pre-training with STFT features without augmentation, the audios in the training data were downsampled to 8kHz then frames generated by applying short-time Fourier transform using a 32 ms Hamming window and an 8 ms hop size. While pre-training using STFT features with augmentation, we first created the first set (set A) of 30 hours of noisy speeches in time domain by adding randomly sampled excerpt from noise recorded in various urban setting from (Wichern et al., 2019) to the 30 hours of clean speech from $si\_tr\_s$. The second set (set B) was created by adding reverberation to the first set using edited scripts from (Maciejewski et al., 2020)( see Figure 2). The two sets of speeches were then downsampled to 8kHz and frames were generated by applying short-time Fourier transform using 32 ms Hamming window and 8 ms shift. Two frames extracted from a similar position in both sets were considered to be augmentations of the clean frame in the same position in the original clean speech and hence constitute a positive pair. For time domain pre-training with

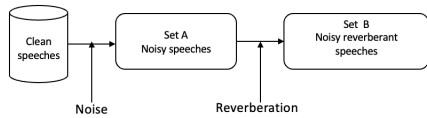

Figure 2: Creating two sets of speeches by adding noise to the first set and noise+reverberation on the second set.

no augmentation, similar to (Subakan et al., 2021b), we divide a given speech in the training set into frames (chunks) of size 250 with 125 overlap between two subsequent frames while for pre-training in time domain with augmentation, we first augment a given audio dataset to create two sets of 30 hour long utterances where one set is augmented with noise and the other has noise plus reverberation as described before. Both sets of speeches are then fragmented into frames (chunks) of size 250 with 125 overlap between two subsequent frames. Frames extracted from similar positions in both

sets are considered augmentations of the clean frame in the same position from the original clean speech, hence forming a positive pair.

**Pre-training configuration**: To train the four variants of pre-trained models, we used the Adam optimiser and the cyclical learning rate (Smith, 2017) with a minimum learning rate of $1e - 4$ and a maximum of $1e - 1$. Each model was trained with a single NVDIA V100 GPU for 1M steps with a batch size of 512 frames.

**Fine-tuning:** For the downstream task of frame partitioning, we do not perform any domain adaptation such as fine-tuning.

**Speech separation**. To evaluate the quality of embeddings generated by pre-trained models on speech separation, we use wsj0-2mix, wsj0-3mix Hershey et al. (2016a),wsj0-4mix, ws0-5mix (Nachmani et al., 2020), Libri5Mix, Libri10Mix (Dovrat et al., 2021). The wsj0-2mix, wsj0-3mix, wsj0-4mix, and ws0-5mix datasets are made of 2, 3, 4, 5 speaker mixtures, respectively, created from the WSJ0 corpus. The datasets are created by exploiting randomly selected gains in order to achieve relative levels between 0 and 5 dB between the 2, 3, 4, 5 speech signals. The datasets are composed of 30 h training, 10 h validation, and 5 h test sets. The training and validation sets share common speakers, which is not the case for test set. Libri5Mix and Libri10Mix are speech mixture composed of 5 and 10 different speakers respectively. The dataset is created from the LibriMix dataset (Cosentino et al., 2020), which was created from LibriSpeech Panayotov et al. (2015). The mixtures are created from clean utterances with no noise with the resulting mixtures having an SNRs that are normally distributed with a mean of 0 dB and a standard deviation of 4.1 dB. These mixtures are created in Dovrat et al. (2021). For all of these datasets, we use the test dataset for speech separation. For each audio in the test dataset, in time-frequency domain we establish frames by applying short-time Fourier transform using 32 ms Hamming window and 8 ms hop size, while in time domain chunks of size 250 with 125 overlap between two subsequent frames are extracted. The frames are then processed by the relevant pre-trained model for embedding generation, e.g., if a pre-trained model was trained using time domain frames it processes time-domain frames to generate embeddings. After embeddings have been generated, we optimize the downstream model according to equation 12 to generate partitions. We set the maximum number of clusters $k = 20$.

**Evaluation metrics**: We used objective metrics of Short-time objective intelligibility (STOI)(Taal et al., 2011), perceptual evaluation of speech quality (PESQ) algorithm (Rix et al., 2001), SI-SNR improvement (SI-SNRi), Signal-to-Distortion Ratio improvement (SDRi), Deep Noise Suppression MOS (DNSMOS) which is a reference-free metric that evaluates perceptual speech quality Reddy et al. (2021) . It is a DNN based model trained on human ratings obtained by using an online framework for listening experiments based on ITU-T P.808. We also use SIG, BAK, OVRL: The non-intrusive speech quality assessment model DNSMOS P.835 (Reddy et al., 2022).

### 6.2 QUALITY OF CLUSTERS

To begin our experiments, we first evaluate how the different frame embeddings resulting from the different pretrained models affect the downstream cluster generation. To evaluate how good the clusters are, we use the graph-based cluster measurement metrics proposed in (Yang & Leskovec, 2012). We are particularly interested in metrics that capture the how well a given partition is separated from the rest i.e., quantifying the number of edges pointing from a given partition to other partitions. A good partition should have few edges pointing outwards. The most relevant metrics for our study being graph modularity and conductance. **Cluster conductance (C)**$= \frac{c_s}{2m_s + c_s}$, if $S$ is a partition, the function C measures how similar the nodes of $S$ are where $m_s$ is the number of edges in $S$ i.e., $m_s = \{(u, v) \in E, u \in S, v \in S\}$ and $c_s$ is the number of edges in the boundary of $S$ i.e. $c_s = \{(u, v) \in E : u \in S, v \notin S\}$. Conductance quantifies the fraction of edges pointing outside a given partition. Quality partitions should have a small conductance value. **Graph modularity ($\mathcal{Q}$)**$= \frac{1}{4}(m_s - E(m_s))$ where $E(m_s)$ is the expected $m_s$. Quality partitions should have high modularity. The results are reported in Table 1. When STFT features with explicit augmentations are used to train the self-supervised model, it generates more quality embeddings that lead to quality clusters as compared to the other three variants of inputs. This variant generates better clusters in all the three test datasets. Notably, time domain with augmentations generates the second-best clusters in all the three datasets. These indicates that for this setup, explicit augmentations are important when training a pre-trained model for speech separation. It is also important to note that STFT trained pre-trained model generates better embeddings that that of time domain suggesting that feature extraction through STFT transform captures more speech separation features as compared to time-domain features.

Table 1: Results of conductance C and modularity Q when using different input configurations and different mixtures. Here the values of C and Q have been multiplied by 100.

| WSJ0-3mix test-dataset | | |
|---|---|---|
| Input Type | C | $Q$ |
| Time domain+augmentation | 15.6 | 86.7 |
| STFT+augmentation | 14.9 | 88.1 |
| Time domain | 17.3 | 81.4 |
| STFT | 16.6 | 83.5 |
| **WSJ0-4mix test-dataset** | | |
| Time domain+augmentation | 16.2 | 85.3 |
| STFT+augmentation | 15.5 | 86.9 |
| Time domain | 18.0 | 79.7 |
| STFT | 17.5 | 82.4 |
| **WSJ0-5mix test-dataset** | | |
| Time domain+augmentation | 19.0 | 76.6 |
| STFT+augmentation | 18.4 | 78.5 |
| Time domain | 20.3 | 75.4 |
| STFT | 19.6 | 75.8 |

## 6.3 EVALUATION ON SPEECH SEPARATION

We evaluated the performance of the proposed technique on source separation using wsj0-2mix, wsj0-3mix wsj0-4mix, ws0-5mix test datasets. Table 2 reports the results based on the evaluation metrics. The quality of partitions has direct relationship with the quality of speech separation. In all the four datasets, the pre-trained model where STFT is contaminated with noise and reverberation registers significantly higher results as compared to the other three. Like the observation in section 6.1, pre-trained model trained with time domain features with augmentations registers the second-best performance. The results show that injecting noise and reverberation in the pre-training increases the portability of the generated features to the speech separation domain. Further, with this setup, STFT based features generate significantly better features as compared to time domain features. It is worth noting that even without domain adaptation, the proposed deep modularization technique can generate quality estimated clean speech signals. We also note that with deep modularity technique, performance drops marginally as the number of sources increases. A direct comparison with other existing speech separation tools is included in Appendix A1, A2.

Table 2: Showing speech separation results when different variants of inputs are used to in the pre-trained model.

| WSJ0-2mix test-dataset | | | | | | | | |
|---|---|---|---|---|---|---|---|---|
| Model | SI-SNRi(↑) | SDRi(↑) | STOI(↑) | PESQ (↑) | DNSMOS (↑) | SIG (↑) | BAK (↑) | OVRL (↑) |
| ConDeepMod(STFT+augmentation) | 21.6 | 20.9 | 0.9069 | 3.98 | 4.05 | 3.98 | 4.11 | 4.01 |
| ConDeepMod(STFT+augmentation) | **22.9** | **22.7** | **0.9346** | **4.04** | **4.17** | **4.11** | **4.23** | **4.16** |
| ConDeepMod(Time domain) | 19.8 | 20.1 | 0.9123 | 3.92 | 3.93 | 3.89 | 3.97 | 3.89 |
| ConDeepMod(STFT) | 21.3 | 21.0 | 0.9323 | 4.01 | 3.97 | 3.88 | 4.08 | 3.94 |
| **WSJ0-3mix test-dataset** | | | | | | | | |
| ConDeepMod(Time domain+augmentation) | 21.2 | 20.7 | 0.9123 | 3.96 | 3.98 | 3.93 | 4.03 | 3.99 |
| ConDeepMod(STFT+augmentation) | **22.1** | **22.4** | 0.9146 | **4.01** | **4.09** | **4.02** | **4.17** | **4.08** |
| ConDeepMod(Time domain) | 19.3 | 18.8 | 0.8790 | 3.88 | 3.85 | 3.83 | 3.90 | 3.87 |
| ConDeepMod(STFT) | 20.9 | 20.6 | **0.9301** | 3.91 | 3.90 | 3.86 | 4.02 | 3.96 |
| **WSJ0-4mix test-dataset** | | | | | | | | |
| ConDeepMod(Time domain+augmentation) | 15.9 | 14.9 | 0.9007 | 3.97 | 3.94 | 3.90 | 3.99 | 3.93 |
| ConDeepMod(STFT+augmentation) | **16.3** | **16.0** | **0.9102** | **4.07** | **4.00** | **3.99** | **4.09** | **4.03** |
| ConDeepMod(Time domain) | 14.6 | 14.7 | 0.8630 | 3.86 | 3.81 | 3.79 | 3.86 | 3.80 |
| ConDeepMod(STFT) | 15.0 | 14.9 | 0.9045 | 3.92 | 3.87 | 3.86 | 3.91 | 3.85 |
| **WSJ0-5mix test-dataset** | | | | | | | | |
| ConDeepMod(Time domain+augmentation) | 13.8 | 14.3 | 0.8787 | 3.94 | 3.91 | 3.85 | 3.90 | 3.87 |
| ConDeepMod(STFT+augmentation) | **14.2** | **14.7** | **0.9089** | **4.003** | **3.95** | **4.90** | **4.04** | **4.001** |
| ConDeepMod(Time domain) | 13.2 | 13.4 | 0.8730 | 3.80 | 3.79 | 3.74 | 3.80 | 3.81 |
| ConDeepMod(STFT) | 14.0 | 13.9 | 0.8745 | 3.86 | 3.80 | 3.77 | 3.85 | 3.83 |

## 7 CONCLUSION

Through experiments, we establish that injecting noise and reverberation in the speech training data helps pre-trained models learn better features tailored for speech separation. This justifies the need to develop a pre-trained model tailored for speech separation rather than using the general audio pre-trained models. Further, based on the proposed model, we establish that working in STFT domain results in higher quality embeddings as compared to time domain features. The proposed speech separation technique based on deep modularization is effective in establishing independent sources contained in a mixture and can work in mixtures with unknown sources.

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

# A APPENDIX

## A.1 COMPARISON WITH OTHER SPEECH SEPARATION TOOLS IN FEW ( $n \leq 3$) SOURCE MIXTURES.

Here, we compare how the proposed technique of speech separation performs as compared to other state of the art speech separation tools. The results are reported in Table 3. In wsj0-2mix, ConDeep-Mod (STFT+augmentation) improves SI-SNRi and SDRi by 0.6 and 0.3 respectively as compared to SepFormer+DM. In wsj0-3mix, the scalability of the proposed technique to high source mixtures is evidence as compared to the other tools. While the performance of SepFormer + DM with regard to SI-SNRi and SDRi drops by 2.8 and 2.7 respectively in the wsj0-3mix dataset when compared to its performance in wsj0-2mix, the performance of ConDeepMod (STFT + augmentation) drops marginally by only 0.7 and 0.3 respectively in the two metrics. This may signal the ability of modularity technique to generalize to mixture with high sources.

Table 3: Comparing the results of the proposed technique with other state of the art speech separation tools.

| Model | SI-SNRi | SDRi |
|---|---|---|
| **WSJ0-2mix test-dataset** | | |
| SepFormer Subakan et al. (2021b) | 20.4 | 20.5 |
| SepFormer+DM Subakan et al. (2021b) | 22.3 | 22.4 |
| Wavesplit Zeghidour & Grangier (2021b) | 21.0 | 21.2 |
| Wavesplit+DM Zeghidour & Grangier (2021b) | 22.2 | 22.3 |
| DeepCASA Liu & Wang (2019b) | 17.7 | 18.0 |
| ConvTasnet Luo & Mesgarani (2019b) | 15.3 | 15.6 |
| ConDeepMod(Time domain+augmentation) | 21.6 | 20.9 |
| ConDeepMod(STFT+augmentation) | **22.9** | **22.7** |
| ConDeepMod(Time domain) | 19.8 | 20.1 |
| ConDeepMod(STFT) | 21.3 | 21.0 |
| **WSJ0-3mix test-dataset** | | |
| SepFormer | 17.6 | 17.9 |
| SepFormer+DM | 19.5 | 19.7 |
| Wavesplit | 17.3 | 17.6 |
| Wavesplit+DM | 17.8 | 18.1 |
| ConvTasnet | 12.7 | 13.1 |
| ConDeepMod(Time domain+augmentation) | 21.2 | 20.7 |
| ConDeepMod(STFT+augmentation) | **22.1** | **22.4** |
| ConDeepMod(Time domain) | 19.3 | 18.8 |
| ConDeepMod(STFT) | 20.9 | 20.6 |

## A.2 COMPARISON WITH OTHER SPEECH SEPARATION TOOLS IN HIGH( $n \geq 5$) SOURCE MIXTURES.

Here, we evaluate the performance of the proposed technique in mixtures with many sources. The results are shown in Table 4. The best performing variant of the proposed technique outperforms the existing tools by 0.4, 0.7, 1.0 and 1.8 when evaluated on wsj0-5mix, Libri5Mix and Libri10Mix dataset on SDRi metric. This shows the proposed technique can scale to high source mixtures and generate quality estimated sources.

## A.3 SELECTING SIMILARITY THRESHOLD $\theta$

Selecting the ideal threshold ($\theta$) when creating adjacency matrix is not trivial. If $\theta$ is high, we risk losing important relationships between frames. On the other hand, selecting low $\theta$ results in a large graph dominated by uninformative edges and increases the clustering time. To select the optimum $\theta$ we conducted experiments with different datasets where we varied the value of $\theta$ and recorded modularity and the number of clusters generated. The graph showing how modularity and number of clusters generated vary when using ws0-5mix test dataset is shown in figure 3. The modularity values in figure 3 have been normalised by multiplying by 100, and values of number of clusters have been normalised by multiplying by 10 for easy visualisation. As can be seen in figure 3, as the similarity increases, modularity increases at the risk of generating a singleton partition. Decreasing

Table 4: Performance of the proposed technique on high source mixtures as compared to other tools that can perform high source mixtures separation.

| Results on the WSJ0-5mix test-dataset | |
|---|---|
| **Model** | **SDRi** |
| ConvTasNet Luo & Mesgarani (2019a) | 6.8 |
| DPRNN Luo et al. (2020) | 8.6 |
| MulCat Nachmani et al. (2020) | 10.6 |
| Hungarian Dovrat et al. (2021) | 13.2 |
| ConDeepMod (Time domain+augmentation) | 13.8 |
| ConDeepMod (STFT+augmentation) | **14.2** |
| ConDeepMod (Time domain) | 13.2 |
| ConDeepMod (STFT) | 14.0 |
| **Libri5Mix test-dataset** | |
| SinkPIT Tachibana (2021) | 9.4 |
| MulCat Nachmani et al. (2020) | 10.8 |
| Hungarian Dovrat et al. (2021) | 12.7 |
| ConDeepMod (Time domain+augmentation) | 13.4 |
| ConDeepMod (STFT+augmentation) | **13.7** |
| ConDeepMod(Time domain) | 12.7 |
| ConDeepMod (STFT) | 13.2 |
| **Libri10Mix test-dataset** | |
| SinkPIT Tachibana (2021) | 6.8 |
| MulCat Nachmani et al. (2020) | 4.8 |
| Hungarian Dovrat et al. (2021) | 7.8 |
| ConDeepMod(Time domain+augmentation) | 9.2 |
| ConDeepMod(STFT+augmentation) | **9.6** |
| ConDeepMod(Time domain) | 7.7 |
| ConDeepMod(STFT) | 8.2 |

the similarity lowers modularity and the risk of generating extra partitions increases. In our case we selected a $\theta = 0.3$. The same threshold is used in all the experiments.

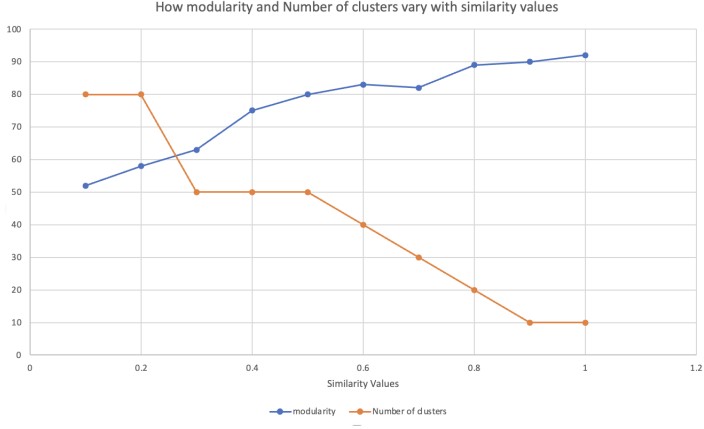

Figure 3: Graph showing how modularity and number of clusters vary as we change the similarity threshold

## A.4 MODEL

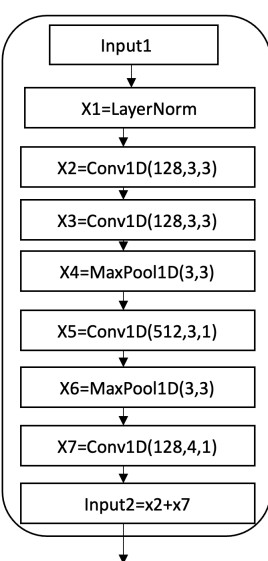

Figure 4: The proposed Encoder, where Conv1D(x,y,z) represents a 1D convolution with $filters = x$, kernel size= $y$ and $strides = z$. MaxPool(x,y) is a 1D maxpooling with $poolsize = x$ and $strides = y$

.

