# OpenReview forum: "Which pre-trained model is effective for speech separation ?"
_ICLR.cc/2024/Conference — Submitted to ICLR 2024_

### Official Review · Reviewer_2QbT · 2023-10-24

**Soundness:** 3 good
**Presentation:** 2 fair
**Contribution:** 3 good
**Rating:** 5
**Confidence:** 5

**Summary:**

The paper investigates using pre-trained models to generate representations for speech separation, hypothesizing that models trained on noisy/reverberant speech will perform better than ones trained on clean speech. They propose a fully unsupervised speech separation technique based on deep modularization that avoids permutation ambiguity. Through experiments, they find that contaminating the pre-training data with noise/reverberation improves performance, as does using STFT features over time-domain, and their proposed technique performs well on separations with varying numbers of speakers.

**Strengths:**

1. The authors introduced contrastive learning and deep modularization networks into the domain of speech separation. They employed models pre-trained on a training dataset to extract features and accomplish speech separation tasks. Experimental results indicate a commendable quality of separation.

2. The authors employed the deep modularization network method to maximize the separation of mixed speech signals. Compared to other separation models, their approach achieves superior results in situations with multiple speakers.

**Weaknesses:**

1. While the authors claim to explore the influence of pre-trained models on the separation task, the paper appears more akin to a two-stage training paradigm leveraging contrastive learning. Conventionally, pre-trained models refer to general-purpose speech models, such as Wav2vec and WavLM.
2. The two core contributions of this paper are the pre-trained model using contrastive learning and the deep modularization network. Firstly, I believe the authors should demonstrate the significance of the pre-trained model for the separation task. The authors displayed in Table 1 the impact of different pre-trained inputs on performance when using a deep modularization network. This perspective might be narrow, as it may be specifically tailored for this network structure. Comparative evaluations with other separation networks like Conv-TasNet and DPRNN are warranted. Secondly, the authors should contrast the use of existing pre-trained models combined with the deep modularization network to underscore the network's significance.
3. The authors' claim of using an unsupervised method seems to be imprecise. During the pre-training phase, the Encoder has already acquired all the single speaker information from the training set. If the approach were genuinely unsupervised, the mixed signals should be used during the pre-training phase.
4. The paper contains some typographical errors and figure reference inaccuracies. For instance, all equations should be followed by punctuation, "equation 3" in Figure 1 should be "equation 2", and in Section 4, "$A_{ij}$ iff" should be corrected to "if". I hope the authors can meticulously address these mistakes.

**Questions:**

1. Given that the authors utilized speech signals with noise and reverberation as inputs for the pre-trained model, can their method separate mixed signals that contain noise or reverberation? For instance, have experiments been conducted on datasets like WHAM! or WHAMR!?
2. I perceive the clustering parameter 'k' as a pivotal hyperparameter. Is it possible to evaluate the model's performance under various 'k' values?
3. Considering that, for example, TF-GridNet can achieve an SI-SNRi=23.4 on WSJ0-2mix, the authors' method of pre-training followed by separation is inherently more intricate. How do the authors rationalize this complexity?
4. The datasets used by the authors have a 100% overlap. Does the deep modularization network proposed by the authors remain viable for data with varying degrees of overlap?

---

> ### Author Response · Authors · 2023-11-12
> **Rebuttal**
>
> Response to  weakness 1(W1): The main argument of the paper is that these general-purpose speech models have failed to generate features that improve speech separation based on [1]. We are investigating whether a standalone pre-trained model tailored for speech separation will result in better features. We investigate these with four variations of pre-trained models trained with the following varying input features.
> •	DFT features +augmentations
> •	DFT features without augmentations
> •	Raw waveform features +Augmentations
> •	Raw waveform features without augmentations
>
> Response to W2: We invite the reviewer to look at the table 4 in the appendix, where we compare the results with the existing technique. Since deep modularization network clusters data based on  what we refer as frame i.e., spectrogram datapoints, it is difficult  to fit this input to existing pre-trained model since their input were not structured this way. However, we acknowledge this is a valid concern. But we rely on justification from [1] that their effectiveness in  speech separation is limited.
>
> Response to W3: The model is trained in Wall Street Journal (WSJ0) corpus dataset, and it is able to generalize to Libri5Mix, Libri10Mix even without domain adaptation technique see table 4 in appendix. We also compare the results with other tools in appendix in table 4.
>
> Response to W4: This well noted. We have corrected based on this observation.
>
> Response to question a(Q1): We use the common speech separation dataset wsj0-2mix, wsj0-3mix ,wsj0-4mix, ws0-5mix,  Libri5Mix and Libri10Mix which has both training, validation, and testing. The training and validation have overlap but testing has no overlap with training and validation I.e., the training and validation sets share common speakers, which is not the case for test set. This explained in pg 8. Furthermore, the models are trained using WSJO dataset but are able to perform well in Libri5Mix and Libri10Mix( see table 4 in the appendix )  without any data adaptation technique. The dataset have been used by our baseline tools so  allows for direct comparison. We did not evaluate the tool in the dataset that have reverberation. However, the performance of the proposed tool in the mixtures which have  reverberation and noise can be explored further.
>
> Response to Q2 : Since our dataset contained mixtures having 10 speakers  we evaluated with k=10,k=15 and k=20 and k=20 provided optimum results so k was set experimentally.
>
> Response to Q4: Our work was not entirely to propose a new speech separation tool, however our key goal was to
> 1.	Establish whether  there is need for a standalone pre-trained model for speech separation where the training dataset is contaminated by noise and reverberation, or the existing general audio pre-trained model suffice.
>
> 2.	Establish the  best type of input ( raw waveform or Fourier based features) are effective for training the pre-trained models tailored for speech separation. The proposed deep modularization is used to avoid the problem of permutation ambiguity during speech separation
>
> 3.	Most tools use supervised technique of speech separation which suffer from the problem of permutation ambiguity. They solve this by using permutation invariant training (PIT).  PIT has a computation complexity of $O(S!)$ with $S$ being number of speakers. The proposed method has a complexity of  (O)(d2n)( see pg 6).
>
> reference
>
> [1] Zili Huang, Shinji Watanabe, Shu Wen Yang, Paola Garc ́ıa, and Sanjeev Khudanpur. Investigating Self-Supervised Learning for Speech Enhancement and Separation. ICASSP, IEEE International Conference on Acoustics, Speech and Signal Processing - Proceedings, 2022-May:6837–6841,2022. ISSN 15206149. doi: 10.1109/ICASSP43922.2022.9746303

---

> > ### Comment · Reviewer_2QbT · 2023-11-19
> > **Concerns about noise and reverberant environments and complexity**
> >
> > Thank you for addressing the concerns I raised in my initial review. Your responses provide a clearer understanding of your methodology and intentions with the research.
> >
> > However, I maintain reservations about the robustness of the proposed method in noisy and reverberant environments. While the results in a clean setting are promising, there remains a significant gap between these conditions and real-world scenarios. Additionally, the conclusions drawn from a clean environment may not necessarily transfer to more complex, real-life situations.
> >
> > Furthermore, for a comprehensive evaluation of the proposed model, a detailed analysis of the computational complexity is crucial. This includes both the pre-training and separation processes. Such an analysis would be beneficial for comparing the proposed model to current state-of-the-art techniques, thereby underscoring its advantages or limitations. I encourage the inclusion of these considerations in your revised paper to strengthen your findings.

---

### Official Review · Reviewer_tTem · 2023-11-01

**Soundness:** 3 good
**Presentation:** 3 good
**Contribution:** 2 fair
**Rating:** 5
**Confidence:** 4

**Summary:**

This paper compares the performance of pre-trained models trained on contaminated speeches versus those trained on clean ones for speech separation tasks. The author evaluates whether training on contaminated audio leads to better downstream performance, and also
investigates the impact of the type of input used for training on the quality of embeddings generated by the pre-trained model.
The key findings of the paper are 1. Pre-trained models trained with noise and reverberation generate significantly better embeddings than those trained with clean datasets. 2. Using short-time Fourier transform (STFT) features is more effective than using time-domain features. In terms of the separation approach, a fully unsupervised technique of speech separation based on deep modularization is proposed.

**Strengths:**

- Claimed by the authors, general audio pre-trained models offer only marginal improvements in speech separation compared to features derived without these models. This limited efficacy might stem from the models being trained on clean audio datasets, rendering them less adept at handling noisy and mixed speech environments. Exploring whether incorporating more diverse, noise-contaminated data into the training could enhance their performance, particularly for tasks like speech separation, presents an intriguing avenue of research.

- A novel speech separation model has been introduced, which utilizes features from pre-trained models within a graph-based framework.

- The model has undergone ablation studies involving scenarios with more than two speakers. These studies have generated considerable interest and enthusiasm among the audience.

**Weaknesses:**

The main weaknesses of the paper lie in several aspects listed as follows:

- Missing Key References: The paper overlooks crucial literature related to the use of noise and reverberation in pre-training, particularly the WavLM-based pre-trained model as discussed in Chen et al. ("Wavlm: Large-scale self-supervised pre-training for full stack speech processing." IEEE Journal of Selected Topics in Signal Processing 16.6 (2022): 1505-1518.). Additionally, relevant studies on speech separation using WavLM, such as Chen et al.'s work ("Speech separation with large-scale self-supervised learning." ICASSP 2023), are not cited. These omissions are significant as these works also concluded the beneficial impact of noise and reverberation.

- Lack of Baseline System: The paper is not able to provide baseline performance for the comparable downstream speech separation model. This absence hinders readers' ability to effectively assess and compare the proposed model's effectiveness.

**Questions:**

- Elaborating more on the limitations of the study, as noted above, is crucial for a comprehensive understanding.
- Incorporating an analysis of run-time factors would significantly enrich the paper, considering the typically high computational costs associated with pre-trained models.
- The rationale behind selecting CONDEEPMOD as the pre-trained model remains ambiguous. The paper would benefit from a more detailed justification of this choice.

---

> ### Author Response · Authors · 2023-11-12
> **Rebuttal**
>
> Reviewer: Missing Key References: The paper overlooks crucial literature related to the use of noise and reverberation in pre-training, particularly the WavLM-based pre-trained model as discussed in Chen et al. ("Wavlm: Large-scale self-supervised pre-training for full stack speech processing." IEEE Journal of Selected Topics in Signal Processing 16.6 (2022): 1505-1518.). Additionally, relevant studies on speech separation using WavLM, such as Chen et al.'s work ("Speech separation with large-scale self-supervised learning." ICASSP 2023), are not cited. These omissions are significant as these works also concluded the beneficial impact of noise and reverberation.
>
> Response: While the papers quoted definitely are related to our work since  WavLM was trained with contaminated speech. However,  our work primary focus is to evaluate  if such training( where input is contaminated) makes it generate better features for speech separation as  compared to pre-trained model such as [1] which are trained using clean speech. We investigate this by training four pre-trained models using constrative learning. The four variations of pre-trained models trained with the following varying input features.
> •	DFT features +augmentations
> •	DFT features without augmentations
> •	Raw waveform features +Augmentations
> •	Raw waveform features without augmentations
>
> Work in IEEE Journal of Selected Topics in Signal Processing 16.6 (2022): 1505-1518.) investigates the various techniques to efficiently integrate the WavLM model in speech separation when faced with limited computation budget, including a low frame rate SSL model training setup and a fine-tuning scheme using only the part of the pre-trained model. It does not look at the quality of features generated based on how the model was trained .
>
> Reviewer : Lack of Baseline System: The paper is not able to provide baseline performance for the comparable downstream speech separation model. This absence hinders readers' ability to effectively assess and compare the proposed model's effectiveness.
>
> Response: We think you missed the appendix section where we compare the results with other baseline tools. We invite you to kindly  look at  table 4 in the appendix, where we discuss baseline systems.
>
> Reviewer : Incorporating an analysis of run-time factors would significantly enrich the paper, considering the typically high computational costs associated with pre-trained models.
>
> Response: Most tools use supervised technique of speech separation which suffer from the problem of permutation ambiguity. They solve this by using permutation invariant training (PIT). PIT has a computation complexity of $O(S!)$ with  $S$ being number of speakers. The proposed method has a complexity of (O)(d^2n)( see pg 6) which is much faster.
>
> Reviewer: The rationale behind selecting CONDEEPMOD as the pre-trained model remains ambiguous. The paper would benefit from a more detailed justification of this choice.
>
> Response: In introduction section pg 2 we justify why we use deep modularization
>
> -to avoid permutation ambiguity problem.
>
> -to avoid challenges of data annotation of supervised learning.
>
> However the key goal of the paper was to evaluate
> 1. Whether contamination of input data makes pre-trained model more suitable for speech separation as compared to that trained without contamination
> 2. Which input features are the best for training the models i.e raw waveform or DFT transform features ?
>
>
> [1]Aaqib Saeed, David Grangier, and Neil Zeghidour. Contrastive learning of general-purpose audio representations. In ICASSP 2021-2021 IEEE International Conference on Acoustics, Speech and Signal Processing (ICASSP), pp. 3875–3879. IEEE, 2021

---

> > ### Comment · Reviewer_tTem · 2023-11-23
> >
> > I really appreciate the author's additional information. However, I keep my original rating of the paper. Thank you.

---

### Official Review · Reviewer_NwoZ · 2023-11-02

**Soundness:** 2 fair
**Presentation:** 2 fair
**Contribution:** 3 good
**Rating:** 5
**Confidence:** 3

**Summary:**

This paper describes a new variant of deep clustering for source separation and compares two different supervision signals for pre-training the encoder (same speaker vs simCLR-like dual augmentations) based on two different representations (STFT vs learned filterbank). These representations are then fed into a graph neural network that is further trained to cluster time-frequency points from the same speaker.

It is evaluated on several (clean, non-reverberant) mixtures of wall street journal sentences (wsj0-2mix, 3mix, 4mix, 5mix) along with other mixtures of LibriSpeech (Libri5Mix, Libri10Mix). The proposed system out-performs a number of systems described up until 2021, including SepFormer, SepFormer+DM, Wavesplit, Wavesplit+DM, and DeepCASA in terms of SI-SNR improvement and SDR improvement.

**Strengths:**

* The approach appears to perform quite well in comparison to existing methods.
* The literature review is very thorough and comprehensive
* The proposed work seems to be well motivated
* The experiments seem to be carried out carefully

**Weaknesses:**

In terms of clarity, I find the paper difficult to understand. One major point of confusion is the use of the term "frame" to, I believe, refer to individual spectrogram points. A frame of a spectrogram is an entire column, so it is not clear whether it is spectrogram points or spectrogram columns that are being clustered. Based on the results, I assume it is points, because clustering columns would not produce sufficient speech enhancement, but, for example, section 6.1 states "we divide a given speech in the training set into frames (chunks) of size 250 with 125 overlap between two subsequent frames." No units are provided for these numbers, but they appear to be columns of spectrograms. This critical detail is very unclear.

Additionally, the title of the paper suggests that existing pre-trained models (e.g., wav2vec 2.0) will be compared for speech separation. The abstract further argues that existing pre-trained models are trained on clean speech and so have trouble representing mixed speech. This line of reasoning is quickly abandoned, however, and a new model trained on a relatively small corpus is introduced instead of a large self-supervised model.

Several different loss functions are introduced, and it is not clear when each one is used.

The results tables from the appendix should be included in the main text because they include comparisons to baseline systems. The current tables need not be included and could be replaced by these.

## Minor comments

The paper contains no paragraph breaks, making it unfriendly to read. The paper would be improved by removing some of the less relevant literature review to make space for proper spacing. Additionally, citations are all \cite{}, which are rendered as "Isik et al. (2016)" even when the entire citation should be in the parentheses. This effect can be achieved by instead using \citep{}, which will be rendered as "(Isik et al., 2016)".

The spacing around | | at the beginning of section 4 is very odd.

**Questions:**

What do you mean by "frame"? And if it is indeed a traditional frame (i.e., spectrogram column), then is separation performing within frames or each frame is assigned to a single source entirely?

---

> ### Author Response · Authors · 2023-11-12
>
> Reviewer: In terms of clarity, I find the paper difficult to understand. One major point of confusion is the use of the term "frame" to, I believe, refer to individual spectrogram points. A frame of a spectrogram is an entire column, so it is not clear whether it is spectrogram points or spectrogram columns that are being clustered. Based on the results, I assume it is points, because clustering columns would not produce sufficient speech enhancement, but, for example, section 6.1 states "we divide a given speech in the training set into frames (chunks) of size 250 with 125 overlap between two subsequent frames." No units are provided for these numbers, but they appear to be columns of spectrograms. This critical detail is very unclear.
>
> Response : Note that we pre-train the models( in our case four models) using raw waveform and DFT transformed raw waveform therefore we adopt a common name frame to refer to datapoints. For raw waveform input, given a time-domain speech signal x ∈ R^T , the signal is processed by an encoder to generate representation h ∈ R^{F ×T} . This is then chunked
> into frames along the time axis to generate L ∈ R^{F ×S×N} . Here N represents the number of frames
> generated.The representation h is  chunked  into frames of size 250 with 125 overlaps between two subsequent frames.. For DFT transformed waveform, from the resulting frequency-domain representations we extract frames of spectrogram (T-F bins). These frames of spectrograms serve as our datapoints.
>
> Reviewer: Additionally, the title of the paper suggests that existing pre-trained models (e.g., wav2vec 2.0) will be compared for speech separation. The abstract further argues that existing pre-trained models are trained on clean speech and so have trouble representing mixed speech. This line of reasoning is quickly abandoned, however, and a new model trained on a relatively small corpus is introduced instead of a large self-supervised model.
> Response: Our argument is that  the effectiveness of  general audio pre-trained model such as wave2vec have been investigated in [1] where they conclude they do not generate quality features for speech separation.  Their experiments establish that the 13 pre-trained models used do not significantly improve feature representations as compared to those of baselines in speech separation.  We therefore investigate if:
> 1.	Adding noise will make pre-trained model develop robust features for speech separation.
> 2.	 Adding noise+reverberation will make pre-trained model develop robust features for speech separation.
> 3.	Does input (DFT based vs raw waveform) features used to train pre-trained model significantly  affect the quality of representation generated by the pre-trained model. Where we have four variants of pre-trained models
> •	DFT features +augmentations
> •	DFT features without augmentations
> •	Raw waveform features +Augmenations
> •	Raw waveform features without augmentations
> With regard to data size, based on our technique of using  spectrogram frames, a 30 hr speech signals generates over 15 millions frames (datapoints)
>
> Reviewer: Several different loss functions are introduced, and it is not clear when each one is used.
>
> Response: We use  only two loss functions one for training pre-trained model(equation2) and one for clustering data points(equation12). Confusion is maybe generated  based on equation 11 which we expand to generate 12.
>
> Reviewer :The results tables from the appendix should be included in the main text because they include comparisons to baseline systems. The current tables need not be included and could be replaced by these.
>
> Response :We tried this, but the strict page limit does not allow it to fit even after removing section 6.2.
>
> Reviewer : The paper contains no paragraph breaks, making it unfriendly to read. The paper would be improved by removing some of the less relevant literature review to make space for proper spacing. Additionally, citations are all \cite{}, which are rendered as "Isik et al. (2016)" even when the entire citation should be in the parentheses. This effect can be achieved by instead using \citep{}, which will be rendered as "(Isik et al., 2016)".
>
> Response :We have corrected citation. Due to strict page limitation, we resorted to exploit as much space as possible.

---

> > ### Comment · Reviewer_NwoZ · 2023-12-04
> > **Response to rebuttal**
> >
> > I would like to thank the authors for their rebuttal. I have read it, along with the other reviews and the authors' responses to them and would like to keep my review and score unchanged.

---

### Official Review · Reviewer_VfQ2 · 2023-11-02

**Soundness:** 2 fair
**Presentation:** 3 good
**Contribution:** 1 poor
**Rating:** 5
**Confidence:** 4

**Summary:**

The paper presents an empirical study on the impact of the input feature choice and augmentation on speech separation as a downstream task. The pre-training is tailored to extract contrastive features.

**Strengths:**

The results presented in the paper reaffirm the established scientific consensus about the role of augmentation and its necessity.
The paper presents a model for source separation as a downstream task.
The idea of implementing source separation using pre-trained embedding extractor is though not novel but valuable.

**Weaknesses:**

While it's always valuable to have new data that supports existing theories and frameworks, it is not providing in an apparent manner novel  insights as it is generally understood in the field that augmentation helps with generalization and specifically in the field of audio enhancement it is a general practice to apply data augmentation.
The chosen downstream task is very much related to the pre-training, e.g., the research on contrastive predictive coding has already shown that using the loss used in the present paper leads to distinctive embeddings for different speakers. Hence, the power of the extracted embeddings doesn't seem to be conclusive from this downstream task.

**Questions:**

I appreciate the thoroughness of your work and how it reaffirms the established scientific consensus. It's always valuable to have new data that supports existing theories and frameworks. While the results presented appear to be in line with what is currently understood in the field, I'm curious to know if there are any aspects of your research that you believe could be built upon or explored further to uncover new insights or if there are any unique implications of your findings that might not be immediately apparent.

---

> ### Author Response · Authors · 2023-11-12
> **Rebuttal**
>
> Reviewer: While it's always valuable to have new data that supports existing theories and frameworks, it is not providing in an apparent manner novel insights as it is generally understood in the field that augmentation helps with generalization and specifically in the field of audio enhancement it is a general practice to apply data augmentation. The chosen downstream task is very much related to the pre-training, e.g., the research on contrastive predictive coding has already shown that using the loss used in the present paper leads to distinctive embeddings for different speakers. Hence, the power of the extracted embeddings doesn't seem to be conclusive from this downstream task.
>
> Response: We  appreciate this comment,  We believe that our key  contributions are:
> We note that most audio pre-trained models such as [1] are trained  without adding noise or reverberation to the audio i.e., without augmentation. Could this be the reason why they generate features that are less fit for speech separation ?We investigate this and also whether type of input used to train a pre-trained model matter. We conclude that :
> 1.	Fourier based features register better results ( when used for training pre-trained model) as compared to time domain features even though Fourier based features don’t capture the phase of the signal hence phase has to be factored in during reconstruction.
>
> 2.	Data contamination is key to a pre-trained model generating quality embeddings for speech separation.
>
> 3.	The ability of the deep modularization to avoid permutation ambiguity problem and scale to mixtures with 10 speakers(Table 4) without significant drop in performance  in the appendix is something we believe is a contribution.
>
> 4.	The demonstration that indeed augmentation does boost the quality of features generated by pre-trained model for speech separation is not trivial.
>
> Reviewer : I appreciate the thoroughness of your work and how it reaffirms the established scientific consensus. It's always valuable to have new data that supports existing theories and frameworks. While the results presented appear to be in line with what is currently understood in the field, I'm curious to know if there are any aspects of your research that you believe could be built upon or explored further to uncover new insights or if there are any unique implications of your findings that might not be immediately apparent.
>
> Response:
> 1.	Why does pre-trained model trained on T-F features generate robust features as compared to raw waveform even though phase is not factored in  T-F features.
> 2.	The Deep modularity proposed is able to scale to mixtures with many speakers with just marginal drop in performance as compared to other state of the art speech separation tool. Could this be due to it avoiding permutation ambiguity problem or the clustering technique is just superior in clustering datapoints?
>
> reference
> [1] Aaqib Saeed, David Grangier, and Neil Zeghidour. Contrastive learning of general-purpose audio representations. In ICASSP 2021-2021 IEEE International Conference on Acoustics, Speech and Signal Processing (ICASSP), pp. 3875–3879. IEEE, 2021

---

### Meta-Review · Area_Chair_YFqx · 2023-12-01

**Metareview:**

In this work, the authors present an experimental investigation to demonstrate that models pre-trained to generate representations for speech separation can be improved when trained on noisy/reverberant speech rather than just clean speech. They also propose an unsupervised speech separation technique based on deep modularization. The proposed experimental evaluation gives some new insights on how to possibly improve the generation of features that are a better fit for speech separation task.

The reviewers highlighted several weaknesses and some of these concerns were adequately addressed by the authors during the rebuttal phase. However, there remain still some unresolved issues, for instance, whether the proposed approach works in real-life conditions and a thorough complexity analysis of the proposed solution. Novelty is also a concern.

**Justification For Why Not Higher Score:**

Lacking crucial experimental evidence. Limited novelty.

**Justification For Why Not Lower Score:**

N/A

---

### Decision · Program_Chairs · 2024-01-16

Reject